# COVID-19 vaccine hesitancy and resistance: Correlates in a nationally representative longitudinal survey of the Australian population

**Ben Edwards** *, **Nicholas Biddle, Matthew Gray, Kate Sollis**

ANU Centre for Social Research and Methods, Australian National University, Melbourne, Australia

* ben.edwards@anu.edu.au

## Abstract

### Background

High levels of vaccination coverage in populations will be required even with vaccines that have high levels of effectiveness to prevent and stop outbreaks of coronavirus. The World Health Organisation has suggested that governments take a proactive response to vaccine hesitancy 'hotspots' based on social and behavioural insights.

### Methods

Representative longitudinal online survey of over 3000 adults from Australia that examines the demographic, attitudinal, political and social attitudes and COVID-19 health behavior correlates of vaccine hesitance and resistance to a COVID-19 vaccine.

### Results

Overall, 59% would definitely get the vaccine, 29% had low levels of hesitancy, 7% had high levels of hesitancy and 6% were resistant. Females, those living in disadvantaged areas, those who reported that risks of COVID-19 was overstated, those who had more populist views and higher levels of religiosity were more likely to be hesitant or resistant while those who had higher levels of household income, those who had higher levels of social distancing, who downloaded the COVID-Safe App, who had more confidence in their state or territory government or confidence in their hospitals, or were more supportive of migration were more likely to intend to get vaccinated.

### Conclusions

Our findings suggest that vaccine hesitancy, which accounts for a significant proportion of the population can be addressed by public health messaging but for a significant minority of the population with strongly held beliefs, alternative policy measures may well be needed to achieve sufficient vaccination coverage to end the pandemic.

**Data Availability Statement:** All data files are available from the Australian Data Archives (DOIs:

10.26193/GG2GE3, 10.26193/HLMZNW, 10.26193/GNEHCQ and 10.26193/ZFGFNE).

**Funding:** The Australian Institute of Health and Welfare (AIHW) provided financial support for the collection of the August ANUpoll data presented in this paper.

**Competing interests:** The authors have declared that no competing interests exist.

## 1 Introduction

For most countries the development of a safe and effective vaccination for COVID-19 is seen as the long-term solution to the COVID-19 pandemic. A critical step in extinguishing the pandemic will be vaccination of a high proportion of the population in the context of increasing misinformation, vaccine hesitancy and lack of trust in science. In this paper we present evidence from a large nationally representative survey of vaccination intentions to a safe and effective COVID-19 vaccine. We document the demographic, attitudinal, political and social attitudes and COVID-19 health behaviour correlates of vaccine hesitance and resistance to a COVID-19 vaccine. We focus on lower and higher levels of hesitancy and resistance to vaccination, as tailored public health information campaigns may well be more effective for those who are less hesitant.

Herd immunity of populations is a product of several factors, the infectivity of the coronavirus [1], the effectiveness of the vaccine and the percentage of the population vaccinated. Estimates of infectivity of the coronavirus suggest that with a 100 per cent effective vaccine, 67 per cent of the population needs to be vaccinated [2] but that vaccination coverage needed varies by infectivity (55 per cent when $R_0 = 2.2$ to 82 per cent when $R_0 = 5.7$ [3]. Simulations suggest that solely relying on a vaccine to extinguish a COVID-19 epidemic would require vaccination coverage of 100 per cent with vaccine effectiveness of 60 per cent [4]. For 80 per cent effectiveness of a vaccine, coverage of 75 per cent of the population would be required [4], a high vaccination rate to ensure eradication or control of the coronavirus in populations.

Research to date suggests that COVID-19 vaccination intentions vary substantially between countries [5]. Vaccine hesitancy (being unsure about getting a vaccine) usually accounts for a more substantial share of the population who will not be vaccinated than vaccine resistance (those who object to vaccines). For instance, in the United States 21 per cent were probably willing and 31 per cent were not willing to have the COVID-19 vaccine [6] while nationally representative surveys in the United Kingdom reported 25–27 per cent were hesitant, and 6–9 per cent resistant [7, 8] and in Canada reported 19 per cent somewhat likely, 9 per cent did not know and 14 per cent unlikely to get the COVID-19 vaccine when available [9]. A cross-national representative survey of over 7,000 participants in seven European countries (Denmark, France, Germany, Italy, Portugal, the Netherlands, and the UK) reported that across these countries 19 per cent were hesitant (not sure) and 7 per cent would not get vaccinated [5]. However, there was substantial variation between countries with vaccine hesitancy 12–28 per cent while resistance ranged from 5–10 per cent [5]. In Australia vaccine hesitancy to a COVID-19 vaccine has been reported at 9 per cent and vaccine resistance 5 per cent however this evidence was from a non-representative online survey [10].

The World Health Organisation's Strategic Advisory Group of Experts (SAGE) have suggested governments take a proactive response to vaccine hesitancy 'hotspots' based on social and behavioural insights [11]. Therefore, nationally representative information is important for an agile response to COVID-19 vaccination. Using a representative longitudinal survey of over 3000 participants from Australia we examine the demographic, attitudinal, political and social attitudes and COVID-19 health behaviour correlates of vaccine hesitance and resistance to a COVID-19 vaccine. We distinguish between those who may get the vaccine but are not sure (hesitant) from those who will definitely get the vaccine because they are usually a large percentage of the population, and unlike vaccine resistant individuals, are likely to be convinced about public health messaging and information about vaccine safety [12]. We also examine those who will not get the vaccine (resistant). Given the Australian government has indicated that the COVID-19 vaccine would be provided free to the population, consistent with previous research on vaccine hesitancy [7, 13] we hypothesise that confidence in

government and science, attitudes towards COVID-19 and adherence with public health messages, conservative and authoritarian political attitudes will be more important than demographic characteristics [7, 8, 13]. We also test whether downloading the COVID-Safe app was related to vaccine hesitancy or resistance.

The remainder of the paper is as follows, Section 2 outlines the methodology including the study design and participants, survey questions and statistical analyses. Section 3 shows the results and Section 4 discusses the results in the context of previous research and their implications.

## 2 Methods

### 2.1 Study design and participants

The primary source of data for this paper is the August ANUpoll, which collected data from 3,061 respondents aged 18 years and over across all eight States/Territories in Australia, and is weighted to have a similar distribution to the Australian population across key demographic and geographic variables. Data for the vast majority of respondents was collected online (94.1 per cent), with a small proportion of respondents enumerated over the phone. A limited number of telephone respondents (17 individuals) completed the survey on the first day of data collection, with a little under half of respondents (1,222) completing the survey on the 11[th] or 12th of August.

The ANUPoll used participants from Life in Australia™ a representative online panel initially recruited using dual-frame landline and mobile random digit dialling [14]. Between October-December 2019, the panel was refreshed with n = 347 panellists being retired and n = 1,810 new panellists being recruited. This recruitment used a Geocoded National Address File sample frame and push-to-web methodology. Only online participants were recruited in order to balance the demographics (the age profile of panel members was older and more educated than that of the Australian population). The recruitment rate for the replenishment was 12.1 per cent.

The contact methodology for offline Life in Australia™ members was an initial SMS (where available), followed by an extended call-cycle over a two-week period. A reminder SMS was also sent in the second week of fieldwork. Taking into account recruitment to the panel, the cumulative response rate for the most recent survey is 7.8 per cent, a slight decline from previous waves of data collection in 2020.

Unless otherwise stated, data in the paper is weighted to population benchmarks. For Life in Australia™, the approach for deriving weights generally consists of the following steps:

1. Compute a base weight for each respondent as the product of two weights:

    a. Their enrolment weight, accounting for the initial chances of selection and subsequent post-stratification to key demographic benchmarks

    b. Their response propensity weight, estimated from enrolment information available for both respondents and non-respondents to the present wave.

2. Adjust the base weights so that they satisfy the latest population benchmarks for several demographic characteristics.

The ethical aspects of the ANUpolls have been approved by the ANU Human Research Ethics Committee (2014/241). Informed consent was provided online or verbally depending on the initial means of recruitment. Data and copies of the questionnaire are available through the Australian Data Archive, see: https://dataverse.ada.edu.au/dataset.xhtml?persistentId=doi%3A10.26193%2FZFGFNE.

## 2.2 Survey questions

**2.2.1 Dependent variable.** Vaccine intention was measured by the following question: 'The next questions ask about your views on a vaccine for COVID-19' and then we ask 'If a safe and effective vaccine for COVID-19 is developed, would you. . .' with the following four response categories, alongside the weighted percentage of respondents:

- Definitely not (5.5 per cent);

- Probably not (7.2 per cent);

- Probably (28.7 per cent); and

- Definitely (58.5 per cent)

Consistent with previous literature on vaccine acceptability, we define those who are definitely not going to get the vaccine as vaccine resistant [7]. High levels of hesitancy was defined as those who would probably not get vaccinated while low levels of vaccinated was categorised as low levels of hesitancy because of the uncertainty given that the decision was about a vaccine would be safe and effective [7].

**2.2.2 Independent variables.** Detailed descriptions of independent variables included in the analyses are provided in the S1 Appendix. Demographic variables included sex, age, indigenous status, born overseas (English speak or non-English speaking country), speaks a language other than English at home, education, socio-economic status of the area, capital or non-capital city, employed, and household income. The state of Victoria was experiencing an outbreak in August 2020 marked by a score of 79.7 on COVID-19 Stringency index compared to 52.3 and we identified participants living in Victoria through a dummy variable [15]. As a point of comparison the COVID-19 Stringency index was 67.1 in the United States and 69.9 in the United Kingdom on 12 August 2020. Health related variables included self-rated health and disability or chronic illness. COVID-19 related variables included been tested for COVID-19, worried about yourself or family or friends contracting COVID-19, extent of social distancing behavior, downloaded the COVID-Safe app and considering that there was too much worry about COVID-19. Political, and social attitudes included voting intentions, populism, authoritarianism, religiosity, attitudes towards immigration, social trust, altruism, confidence in Federal government, state government or hospital and health system.

## 2.3 Statistical analyses

We estimated an ordinal probit model using oprobit command in STATA 15.1. Given the large number of independent variables we estimated several models. Model 1 included demographic variables from survey respondents who completed the August 2020 wave of data collection and who had complete vaccination intention data and demographic characteristics (S1 Fig). Model 2 included demographic and health variables with a measure of disability from the February 2020 ANU Poll. Model 3 included demographic and COVID-19 related variables from April and May 2020 ANUPoll. Model 4 included demographic and political and social attitudes from February and April 2020 ANUPoll. To understand the relative importance of the variables included in the models, Model 5 included demographic variables and statistically significant variables ($p < 0.05$) from models 2–4 including variables from February, April and May ANUPoll. There was complete vaccine intention information for 3,052 participants. The number of participants in each of the models varied depending on the rate of survey completion in other ANUPolls and missing data for particular variables in the survey (S1 Fig). We weighted to population benchmarks in all analyses to account for survey design and non-response.

## 3 Results

### 3.1 Vaccine hesitancy and resistance

Almost three-in-five Australians (58.5 per cent) would definitely get the vaccine. We divided vaccine hesitancy into two levels. Low levels of vaccine hesitancy were those who indicated they were likely to get the vaccine but not certain (28.7 per cent) and high levels of vaccine hesitancy those who will probably not get the vaccine (7.2 per cent). We defined those who were resistant as those who indicated that they were definitely not going to get the vaccine (5.5. per cent).

### 3.2 Correlates of vaccine hesitancy and resistance

Descriptive statistics for covariates in the statistical modelling can be found in the S1 Appendix. Table 1 shows the marginal effects from model 1, which included demographic, socioeconomic and geographic variables.

Females were less likely than males to intend to get the vaccine, and more likely to be hesitant and resistant. Those who were older (55–64, 65–74 and those over 75 years) were less likely to be resistant or hesitant and more likely to intend to get vaccinated when it became

**Table 1. Demographic correlates of vaccine resistance and hesitancy, marginal effects.**

| Explanatory variables | Resistant | | Hesitant -High | | Hesitant—Low | | Likely | |
|---|---|---|---|---|---|---|---|---|
| | Marginal effect | Signif. | Marginal effect | Signif. | Marginal effect | Signif. | Marginal effect | Signif. |
| Victoria | -0.003 | | -0.003 | | -0.006 | | 0.012 | |
| Female | 0.011 | * | 0.010 | * | 0.021 | * | -0.042 | * |
| Aged 18 to 24 years | -0.013 | | -0.012 | | -0.026 | | 0.052 | |
| Aged 25 to 34 years | 0.006 | | 0.005 | | 0.010 | | -0.021 | |
| Aged 45 to 54 years | 0.007 | | 0.006 | | 0.011 | | -0.025 | |
| Aged 55 to 64 years | -0.021 | ** | -0.020 | ** | -0.047 | ** | 0.089 | ** |
| Aged 65 to 74 years | -0.030 | *** | -0.030 | *** | -0.075 | *** | 0.134 | *** |
| Aged 75 years plus | -0.038 | *** | -0.041 | *** | -0.112 | *** | 0.191 | *** |
| Indigenous | 0.003 | | 0.003 | | 0.006 | | -0.013 | |
| Born overseas in a main English speaking country | 0.009 | | 0.007 | | 0.013 | | -0.029 | |
| Born overseas in a non-English speaking country | 0.003 | | 0.003 | | 0.005 | | -0.011 | |
| Speaks a language other than English at home | 0.015 | | 0.012 | | 0.022 | | -0.049 | |
| Not completed Year 12 or post-school qualification | 0.009 | | 0.008 | | 0.014 | | -0.031 | |
| Has a post graduate degree | -0.024 | ** | -0.024 | ** | -0.056 | ** | 0.105 | ** |
| Has an undergraduate degree | -0.019 | * | -0.018 | ** | -0.041 | ** | 0.079 | ** |
| Certificate III/IV, Diploma or Associate Degree | 0.001 | | 0.001 | | 0.002 | | -0.005 | |
| Lives most disadvantaged areas (1st quintile) | 0.024 | * | 0.019 | ** | 0.032 | * | -0.075 | * |
| Lives next most disadvantaged areas (2nd quintile) | 0.001 | | 0.002 | | 0.003 | | -0.006 | |
| Lives in next most advantaged areas (4th quintile) | 0.022 | | 0.017 | | 0.029 | | -0.068 | * |
| Lives in the most advantaged areas (5th quintile) | 0.002 | | 0.002 | | 0.004 | | -0.008 | |
| Lives in a non-capital city | 0.009 | | 0.007 | | 0.013 | | -0.029 | |
| Employed | 0.001 | | 0.001 | | 0.001 | | -0.002 | |
| Household income | -.00003 | *** | -.00003 | *** | -.00005 | *** | 0.0001 | *** |
| Proportion | 0.051 | | 0.070 | | 0.298 | | 0.593 | |

Source: ANUpoll, April, May and August 2020.

Notes: Ordered probit model. N = 2,717. Base case is estimated from Victorian, female, 35–44 year old, non-indigenous, Australian born, speaks English at home, Year 12 education, 3rd SEIFA neighbourhood quintile, capital city, not employed and mean household income at the mean ($670.81) Marginal effects that are statistically significant at the 1 per cent level of significance are labelled ***; those significant at the 5 per cent level of significance are labelled **, and those significant at the 10 per cent level of significance are labelled *.

available than those aged 35–44 years old. Compared to those had Year 12 only, those with an undergraduate or postgraduate university degree were less likely to be resistant or hesitant and more likely to intend to be vaccinated.

There were neighbourhood differences, those living in the 4th most disadvantaged quintile of disadvantage were less likely to intend to get vaccinated when compared to those living in the 3rd quintile. Individuals living in households with more household income were less likely to be resistant or hesitant and more likely to intend to get vaccinated. All other demographic variables were not statistically significant.

Health and disability status were not associated with vaccine intentions (model 2, see S1 Appendix).

In model 3 COVID-related variables were included as well as demographic variables (see S1 Appendix). Contrary to other research on the likelihood of getting vaccinated to COVID-19 [13] individual concerns or concerns about relatives or friends contracting COVID-19 were not related to vaccination intentions. Being tested for coronavirus was also not related to vaccination intentions.

People who reported greater levels of social distancing behaviour were less likely to be resistant and more likely to intend to get vaccinated. Similarly, those that had downloaded the COVID-Safe App were less likely to be resistant (-3.0 percentage points lower) or hesitant (high: -2.7 percentage points and low: -5.1 percentage points) and more likely to intend to get vaccinated (+10.8 percentage points). Those who thought too much fuss had been made about COVID-19 were more likely to be resistant (8.1 percentage points) or have high levels of hesitancy (4.2 percentage points) and less likely to intend to get vaccinated (-14.9 percentage points less likely).

In Model 4 we added political and social attitudes to demographic characteristics (see S1 Appendix). There were no statistically significant differences by levels of social trust, altruism or support for authoritarianism. Compared to those who voted for the Coalition, those who voted for Labor were less likely to be resistant (-1.6%, $p < 0.10$), hesitant (high: -1.5%; low: -3.2%) and more likely to intend to get vaccinated (6.3%, $p = 0.05$).

Those who had confidence in their state or territory government or in their hospitals and health system were less likely to be resistant (-3.4 and -4.4 percentage points respectively) or have high levels of hesitancy (-2.8 and -3.4 percentage points respectively) or have low levels of hesitancy (-4.8 and -5.2 percentage points respectively) and more likely to intend to get vaccinated (11.1% and 13.0% respectively).

Those who were more religious were less likely to intend to get vaccinated. People who had more populist views were more likely to be resistant or hesitant (at high or low levels) and less likely to intend to get vaccinated. Finally, those who were more likely to support migration were less likely to be resistant and more likely to intend to get vaccinated.

The final model included statistically significant variables from models 2 to 4 and demographic variables (Table 2). Females were less likely to intend to get vaccinated while those aged 55 and over were more likely to intend get vaccinated. Those with higher household income were less likely to be resistant and more likely to intend to get vaccinated. Differences by levels of education observed in model 1 were explained by other variables. Neighbourhood differences by socio-economic index for areas were evident with those living in the most disadvantaged area more likely to be resistant or hesitant (at high levels) and less likely to intend to get vaccinated.

People exercising greater levels of social distancing were less likely to be resistant and more likely to be vaccinated. Those who downloaded the COVID-Safe App were less likely to be resistant or hesitant and more likely to intend to get vaccinated. Those who reported too much fuss had been made about COVID-19 were still more likely to be resistant or hesitant and less likely to intend to get vaccinated.

**Table 2. Correlates of vaccine resistance and hesitancy–final model, marginal effects.**

| Explanatory variables | Resistant | | Hesitant—High | | Hesitant—Low | | Likely | |
|---|---|---|---|---|---|---|---|---|
| | Marginal effect | Sig | Marginal effect | Sig | Marginal effect | Sig | Marginal effect | Sig |
| Victoria | 0.023 | | 0.020 | * | 0.036 | ** | -0.079 | * |
| Female | 0.018 | * | 0.016 | ** | 0.027 | * | -0.061 | ** |
| Aged 18 to 24 years | -0.024 | | -0.022 | | -0.039 | | 0.085 | |
| Aged 25 to 34 years | 0.001 | | 0.000 | | 0.001 | | -0.002 | |
| Aged 45 to 54 years | 0.005 | | 0.004 | | 0.005 | | -0.013 | |
| Aged 55 to 64 years | -0.030 | ** | -0.028 | ** | -0.052 | ** | 0.110 | ** |
| Aged 65 to 74 years | -0.039 | ** | -0.037 | *** | -0.077 | *** | 0.153 | *** |
| Aged 75 years plus | -0.049 | ** | -0.050 | *** | -0.116 | *** | 0.216 | *** |
| Indigenous | -0.010 | | -0.008 | | -0.013 | | 0.032 | |
| Born overseas in a main English speaking country | 0.019 | | 0.014 | | 0.018 | | -0.050 | |
| Born overseas in a non-English speaking country | 0.000 | | 0.000 | | 0.000 | | -0.001 | |
| Speaks a language other than English at home | 0.019 | | 0.014 | | 0.018 | | -0.052 | |
| Not completed Year 12 or post-school qualification | 0.014 | | 0.011 | | 0.014 | | -0.038 | |
| Has a post graduate degree | -0.022 | | -0.019 | | -0.034 | | 0.075 | |
| Has an undergraduate degree | -0.018 | | -0.016 | | -0.027 | | 0.061 | |
| Has a Certificate III/IV, Diploma or Associate Degree | 0.014 | | 0.011 | | 0.014 | | -0.040 | |
| Lives most disadvantaged areas (1st quintile) | 0.033 | * | 0.023 | * | 0.027 | | -0.084 | * |
| Lives next most disadvantaged areas (2nd quintile) | 0.001 | | 0.001 | | 0.001 | | -0.002 | |
| Lives in next most advantaged areas (4th quintile) | 0.025 | | 0.018 | | 0.022 | | -0.066 | |
| Lives in the most advantaged areas (5th quintile) | 0.010 | | 0.008 | | 0.011 | | -0.029 | |
| Lives in a non-capital city | 0.006 | | 0.005 | | 0.007 | | -0.018 | |
| Employed | -0.004 | | -0.003 | | -0.004 | | 0.011 | |
| Household income | -.00003 | ** | -.00002 | *** | -.00004 | ** | .00009 | *** |
| Too much fuss made about COVID-19 | 0.059 | ** | 0.038 | *** | 0.037 | ** | -0.135 | *** |
| Social distancing behaviour | -0.023 | ** | -0.018 | *** | -0.027 | *** | 0.069 | ** |
| Downloaded the COVID-Safe App | -0.031 | ** | -0.028 | *** | -0.053 | *** | 0.112 | *** |
| Voting intention: Labor | -0.022 | * | -0.019 | ** | -0.034 | ** | 0.076 | ** |
| Voting intention: Greens | -0.004 | | -0.004 | | -0.005 | | 0.013 | |
| Voting intention: Other | -0.020 | | -0.017 | | -0.030 | | 0.067 | |
| Voting intention: Don't know | 0.005 | | 0.004 | | 0.005 | | -0.013 | |
| Confident in state or territory government | -0.029 | ** | -0.021 | ** | -0.025 | * | 0.021 | |
| Confident in hospitals and health system | -0.044 | ** | -0.030 | *** | -0.032 | * | 0.075 | ** |
| Support for migration | -0.005 | | -0.003 | | -0.004 | | 0.106 | *** |
| Populism | 0.003 | * | 0.002 | ** | 0.004 | ** | -0.009 | ** |
| Religiosity | 0.003 | * | 0.002 | * | 0.004 | * | -0.009 | * |
| Proportion | 0.070 | | 0.088 | | 0.355 | | 0.487 | |

Source: ANUpoll, April, May and August 2020.

Notes: Ordinal probit model. N = 2,261. Base case is estimated from Victorian, female, 35–44 year old, non-indigenous, Australian born, speaks English at home, year 12 education, 3rd SEIFA neighbourhood quintile, capital city, not employed and mean household income at the mean ($670.81), Coalition voter, has confidence in their state/territory government, has confidence in hospitals and health system. Other variables estimated at the sample means (social distancing behaviour, support for migration, populism, religiosity). Coefficients that are statistically significant at the 1 per cent level of significance are labelled ***; those significant at the 5 per cent level of significance are labelled **, and those significant at the 10 per cent level of significance are labelled *.

Those who had confidence in their state or territory government or confident in their hospitals and health system were less likely to be resistant or hesitant (at high or low levels) and more likely to intend to get vaccinated. People who were more supportive of migration were more likely to get vaccinated.

Those who reported more populist views were more likely to be resistant or hesitant (at high or low levels) and less likely to intend to get vaccinated. This pattern of results was also evident in terms of level of religiosity although this finding should be treated with caution as there was a higher level of uncertainty around the estimates (statistical significance was only at the 90% level). Similarly, those living in Victoria, with by far the highest numbers of COVID-19 infections in Australia in August 2020, were significantly less likely than Australians from other states to get vaccinated but this was not evident in other statistical modelling, bivariate analyses and should not be considered a robust finding.

## 4 Discussion

In August 2020 36 per cent of Australians are hesitant and 6 per cent resistant to being vaccinated with a safe and effective vaccine for COVID-19 if one was available. Given previous research suggests that the factors associated with vaccine resistance might be different to vaccine hesitancy, we examined demographic, health, COVID-19 related health behaviour and attitudes, and political and social attitudinal correlates.

Many factors were associated with vaccine resistance and hesitancy. Consistent with previous research, females, those with lower levels of household income and living in disadvantaged areas were associated with increased likelihood of vaccine resistance or hesitancy [7, 13]. However, in contrast to previous research, younger people were not less likely to intend to get vaccinated than those aged 35–44 years.

Less adherence to COVID-19 health behaviours was consistently associated with lower likelihood of being vaccine resistant or hesitant (social distancing and downloading the COVID-Safe app) and these were strongly related to vaccine intentions [13]. For example, downloading the COVID-Safe app was associated with an increase of 11 percentage points in the likelihood of being vaccinated. Similarly, there was a seven-percentage point increase in the likelihood of intending to be vaccinated for COVID-19 if an individual moved from the 16th percentile in terms of social distancing to the 50th percentile (a one standard deviation increase). Given that many governments have tracking surveys about social distancing, this information could be used to support targeted campaigns to encourage vaccination in areas of low social distancing.

Consistent with other studies of COVID-19 several variables that could be considered associated with broader societal dissatisfaction and anti-establishment sentiments were also associated with vaccine resistance or hesitancy. Specifically in this study, attitudes about too much fuss being made about COVID-19, lack of confidence in state or territory government, and having more populist sentiments [7, 16]. Other studies have also independently reported that religious beliefs associated with declining a COVID-19 vaccine [13].

For many with low levels of hesitancy, providing information about the safety and efficacy of the COVID-19 vaccine will be critical as other studies have highlight this as important hesitancy [13, 17]. Our analyses suggest that those with resistance or hesitancy are likely to lack trust in those providing health services (e.g. state governments or health systems) and therefore the misinformation about the COVID-19 vaccine that will occur may be even more effective for these groups. Pre-emptively using cognitive inoculation techniques and pre-bunking techniques [18] to actively work against the likely misinformation that will be generated in the development and preparation phase of a vaccine for COVID-19 will be important but probably requires more targeted and nuanced public health messages by trusted members of the community including doctors, journalists and politicians. Other strategies to engender trust among the population could include enlisting community and cultural leaders to assist in the development and spread of information [19].

Those with higher vaccine resistance or hesitancy are more likely to have a set of strongly held beliefs, a lack of trust in those responsible for health (state or territory government and hospitals or health systems) and lower levels of compliance with public health advice for COVID-19 (e.g. lower levels of social distancing, not downloading the COVID-Safe App). If large scale surveys collect information on the extent of compliance with health advice, further nuanced targeting could be employed. Beyond more sophisticated and nuanced public health messaging, it should be noted that a systematic review of research on compulsory vaccination policies suggests that the majority of the population supports these programs [18]. However, none of the studies in the systematic review were conducted during a pandemic where civil liberties were restricted due to lockdowns [19] and a staged approach may well be more proportional [20].

While we used a rich set of variables to predict vaccine intentions some factors were not collected in our surveys. In particular, we did not collect information about concerns about vaccine safety which may be important determinant of vaccine hesitancy and may also explain why females were more likely to be hesitant or resistant than males. Another robust correlate in other studies of vaccine intentions is previous vaccination such as for influenza [13]. Vaccine safety and public discourse will likely be particularly important as media and public scrutiny of vaccine trials is unprecedented for COVID-19 vaccines and further monitoring of public sentiment [7].

Another issue that we did not address in this paper is if there are initial shortages of a vaccine how would these be distributed? A survey experiment using these participants suggests that essential health workers, those with a health condition, those in areas with high levels of COVID-19 and with caring responsibilities should get priority [21]. Moreover, vaccine intentions of respondents did not change these priorities.

## 5 Conclusion

Given that over 75 per cent of the population are likely to need to be vaccinated with a highly effective vaccine to extinguish the epidemic [4] our findings that only 59 per cent of Australians will definitely get vaccinated is sobering and suggests that as noted by WHO [11], proactive measures need to be adopted by countries to encourage vaccination in the community. Our findings suggest that vaccine hesitancy, which accounts for a further significant proportion of the population, and can be addressed by public health messaging. However, for a significant minority of the population with strongly held beliefs that are the likely drivers of vaccination intentions, alternative policy measures may well be needed to achieve sufficient vaccination coverage.

## Supporting information

**S1 Fig.**
(TIF)

**S1 Appendix.**
(DOCX)

## Author Contributions

**Conceptualization:** Ben Edwards, Nicholas Biddle, Matthew Gray.

**Formal analysis:** Ben Edwards.

**Funding acquisition:** Nicholas Biddle.

**Methodology:** Ben Edwards, Nicholas Biddle.

**Writing – original draft:** Ben Edwards.

**Writing – review & editing:** Nicholas Biddle, Matthew Gray, Kate Sollis.

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
