## [Decision Letter · Decision Letter 0]

24 Dec 2020

PONE-D-20-37174

COVID-19 vaccine hesitancy and resistance: Correlates in a nationally representative longitudinal survey of the Australian population

PLOS ONE

Dear Dr. Ben,

Thank you for submitting your manuscript to PLOS ONE. After careful consideration, we feel that it has merit but does not fully meet PLOS ONE’s publication criteria as it currently stands. Therefore, we invite you to submit a revised version of the manuscript that addresses the points raised during the review process.

We look forward to receiving your revised manuscript.

Kind regards,

Francesco Di Gennaro

Academic Editor

PLOS ONE

Journal Requirements:

2. In your Methods section, please provide additional information about the methodology used. Please clarify how participants were selected and recruited in the original survey; and whether you applied additional eligibility criteria for inclusion in this analysis (please provide a participant flowchart) .

Moreover, please ensure that all variables are defined in the Methods section, and that it is clear how these were coded for your analysis.

Finally, please include additional information regarding the survey or questionnaire used in the study and ensure that you have provided sufficient details that others could replicate the analyses. For instance, if you developed a questionnaire as part of this study and it is not under a copyright more restrictive than CC-BY, please include a copy, in both the original language and English, as Supporting Information.

Moreover, please include more details on how the questionnaire was pre-tested, and whether it was validated.

3. Please provide additional details regarding participant consent.

In the ethics statement in the Methods and online submission information, please ensure that you have specified (i) whether consent was informed and (ii) what type you obtained (for instance, written or verbal).

If your study included minors, state whether you obtained consent from parents or guardians.

If the need for consent was waived by the ethics committee, please include this information.

4. Thank you for stating the following after the Conclusion Section of your manuscript:

'Funding

The Australian Institute of Health and Welfare (AIHW) provided financial support for the analysis of the August ANUpoll data presented in this paper.'

'The survey data used in this paper were collected with support from the Australian Institute of Health and Welfare.  AIHW did not have any input to the manuscript.'

5. Please include captions for your Supporting Information files at the end of your manuscript, and update any in-text citations to match accordingly. Please see our Supporting Information guidelines for more information: http://journals.plos.org/plosone/s/supporting-information

Additional Editor Comments:

dear authors follow reviewer suggestion to improve your paper

Reviewers' comments:

Reviewer's Responses to Questions

**Comments to the Author**

1. Is the manuscript technically sound, and do the data support the conclusions?

Reviewer #1: Yes

Reviewer #2: Yes

2. Has the statistical analysis been performed appropriately and rigorously? 

Reviewer #1: Yes

Reviewer #2: Yes

3. Have the authors made all data underlying the findings in their manuscript fully available?

Reviewer #1: Yes

Reviewer #2: Yes

4. Is the manuscript presented in an intelligible fashion and written in standard English?

Reviewer #1: Yes

Reviewer #2: Yes

5. Review Comments to the Author

Reviewer #1: Authors wrote an article on a top topic. Hesitancy and resistance can stop the progress to COVID burden control

Well done

Only some minor suggestions

1- Introduction: delete line 55-68-84-102. Update data on COVID bruden worldwide at the day of resumbission

2.Methods and results No comment

3. Discussion: add some future perspectives as take home message to improve and reduce the hesitancy and resistance

and how is the role of doctors, journalist and politicians to improve it

The role of safety is also important (see and cite The efficacy and safety of influenza vaccination in older people: An umbrella review of evidence from meta-analyses of both observational and randomized controlled studies. Ageing Res Rev. 2020 Sep;62:101118. doi: 10.1016/j.arr.2020.101118. Epub 2020 Jun 18. PMID: 32565328.)

Reviewer #2: The authors have conducted a survey to identify what the potential uptake will be for the COVID-19 vaccines and provided information on the demographics of the population who are most and least likely to be vaccinated. This is an important publication. I think that it might be useful to provide examples of how trust could be built for this vaccine/or has been built for other vaccines. Likewise despite the desperate situation that COVID-19 has put us in it might also be good to comment that it affords a lesson to develop new ways to reach difficult to vaccinate communities.

6. PLOS authors have the option to publish the peer review history of their article (what does this mean?). If published, this will include your full peer review and any attached files.

Reviewer #1: No

Reviewer #2: No

---

## [Author Response · Author response to Decision Letter 0]

10 Feb 2021

Manuscript title: COVID-19 vaccine hesitancy and resistance: Correlates in a nationally representative longitudinal survey of the Australian population 

Manuscript ID: PONE-D-20-37174

Response to Editor and Reviewer comments

Please find our specific responses to comments from the Editor and Reviewers. Thank you for your constructive comments and reviews, detailed responses are below.

Editor comments

"Please ensure that your manuscript meets PLOS ONE's style requirements, including those for file naming. The PLOS ONE style templates can be found at

https://journals.plos.org/plosone/s/file?id=ba62/PLOSOne_formatting_sample_title_authors_affiliations.pdf"

We have reviewed PLOS ONE’s style requirements and edited the manuscripts appropriately.

"In your Methods section, please provide additional information about the methodology used. Please clarify how participants were selected and recruited in the original survey; and whether you applied additional eligibility criteria for inclusion in this analysis (please provide a participant flowchart) . Moreover, please ensure that all variables are defined in the Methods section, and that it is clear how these were coded for your analysis."

We have added additional information about how participants were selected and recruited in the original survey, and included a reference to more detailed recruitment and methodological information in the statistical analyses section:

 Model 1 included demographic variables from survey respondents who completed the August 2020 wave of data collection and who had complete vaccination intention data and demographic characteristics (Fig S1). Model 2 included demographic and health variables with a measure of disability from the February 2020 ANU Poll. Model 3 included demographic and COVID-19 related variables from April and May 2020 ANUPoll. Model 4 included demographic and political and social attitudes from February and April 2020 ANUPoll. To understand the relative importance of the variables included in the models, Model 5 included demographic variables and statistically significant variables (p<0.05) from models 2-4 including variables from February, April and May ANUPoll. There was complete vaccine intention information for 3,052 participants. The number of participants in each of the models varied depending on the rate of survey completion in other ANUPolls and missing data for particular variables in the survey (Fig S1). 

See: Lines 216-228 page 8

In terms of eligibility criterion for inclusion in the analysis, we have clarified this using a participant flowchart.

There are many variables included in our models, descriptions of all variables are in the Appendix that is referred to the Methods section. 

We have also added more information on the coding of variables for our analysis.

"Finally, please include additional information regarding the survey or questionnaire used in the study and ensure that you have provided sufficient details that others could replicate the analyses. For instance, if you developed a questionnaire as part of this study and it is not under a copyright more restrictive than CC-BY, please include a copy, in both the original language and English, as Supporting Information.

Moreover, please include more details on how the questionnaire was pre-tested, and whether it was validated."

Line 174, page 7 We have provided a link to the questionnaire which can be downloaded, see

Data and copies of the questionnaire are available through the Australian Data Archive. 

"In the ethics statement in the Methods and online submission information, please ensure that you have specified (i) whether consent was informed and (ii) what type you obtained (for instance, written or verbal).

If your study included minors, state whether you obtained consent from parents or guardians.

If the need for consent was waived by the ethics committee, please include this information."

Line 173-174, page 7 This has been updated: “Informed consent was provided online or verbally depending on the initial means of recruitment.”

"Thank you for stating the following after the Conclusion Section of your manuscript:

'Funding

The Australian Institute of Health and Welfare (AIHW) provided financial support for the analysis of the August ANUpoll data presented in this paper.'

'The survey data used in this paper were collected with support from the Australian Institute of Health and Welfare. AIHW did not have any input to the manuscript.'

b. Please include your amended statements within your cover letter; we will change the online submission form on your behalf."

We have provided this information in the cover letter

Response to reviewers:

"Introduction: delete line 55-68-84-102. Update data on COVID bruden worldwide at the day of resumbission"

Deletions of lines 55-68-84-102 have been undertaken 

Where appropriate we updated the data on COVID burden 

We have also updated the references as some working papers have now been published in journals. 

"Discussion: add some future perspectives as take home message to improve and reduce the hesitancy and resistance and how is the role of doctors, journalist and politicians to improve it

The role of safety is also important (see and cite The efficacy and safety of influenza vaccination in older people: An umbrella review of evidence from meta-analyses of both observational and randomized controlled studies. Ageing Res Rev. 2020 Sep;62:101118. doi: 10.1016/j.arr.2020.101118. Epub 2020 Jun 18. PMID: 32565328.)"

Line 349, page 13 Thank you for your suggestion, we had included in our discussion the following but now highlight that doctors, journalists and politicians have a role (changes in bold) :

Pre-emptively using cognitive inoculation techniques and pre-bunking techniques [16] to actively work against the likely misinformation that will be generated in the development and preparation phase of a vaccine for COVID-19 will be important but probably requires more targeted and nuanced public health messages by trusted members of the community including doctors, journalists and politicians.

Line 342, page 13 We agree that safety is important, thank you for your suggested reference. This has now been cited on page 13

"The authors have conducted a survey to identify what the potential uptake will be for the COVID-19 vaccines and provided information on the demographics of the population who are most and least likely to be vaccinated. This is an important publication. I think that it might be useful to provide examples of how trust could be built for this vaccine/or has been built for other vaccines. Likewise despite the desperate situation that COVID-19 has put us in it might also be good to comment that it affords a lesson to develop new ways to reach difficult to vaccinate communities."

Lines 349-351, page 13 Thank you for your positive review, we appreciate your suggestions.

We have added the following to the discussion:

Other strategies to engender trust among the population could include enlisting community and cultural leaders to assist in the development and spread of information [19].

---

## [Decision Letter · Decision Letter 1]

8 Mar 2021

COVID-19 vaccine hesitancy and resistance: Correlates in a nationally representative longitudinal survey of the Australian population

PONE-D-20-37174R1

Dear Dr. Ben Edwards,

We’re pleased to inform you that your manuscript has been judged scientifically suitable for publication and will be formally accepted for publication once it meets all outstanding technical requirements.

Kind regards,

Francesco Di Gennaro

Academic Editor

PLOS ONE

Additional Editor Comments (optional):

dear authors congratulations

Reviewers' comments:

Reviewer's Responses to Questions

**Comments to the Author**

1. If the authors have adequately addressed your comments raised in a previous round of review and you feel that this manuscript is now acceptable for publication, you may indicate that here to bypass the “Comments to the Author” section, enter your conflict of interest statement in the “Confidential to Editor” section, and submit your "Accept" recommendation.

Reviewer #1: All comments have been addressed

2. Is the manuscript technically sound, and do the data support the conclusions?

Reviewer #1: Yes

3. Has the statistical analysis been performed appropriately and rigorously? 

Reviewer #1: Yes

4. Have the authors made all data underlying the findings in their manuscript fully available?

Reviewer #1: Yes

5. Is the manuscript presented in an intelligible fashion and written in standard English?

Reviewer #1: Yes

6. Review Comments to the Author

Reviewer #1: Authors improved thier manuscript. They wrote an very interesting paper that now can be publish.

7. PLOS authors have the option to publish the peer review history of their article (what does this mean?). If published, this will include your full peer review and any attached files.

Reviewer #1: No

---

## [Editor Report · Acceptance letter]

10 Mar 2021

PONE-D-20-37174R1 

COVID-19 vaccine hesitancy and resistance: Correlates in a nationally representative longitudinal survey of the Australian population 

Dear Dr. Edwards:

I'm pleased to inform you that your manuscript has been deemed suitable for publication in PLOS ONE. Congratulations! Your manuscript is now with our production department. 

Kind regards, 

on behalf of

Dr. Francesco Di Gennaro 

Academic Editor

PLOS ONE